# Epidemiological Analysis and Genetic Characterization of Parvovirus in Ducks in Northern Vietnam Reveal Evidence of Recombination

**DOI:** 10.3390/ani12202846

**Published:** 2022-10-19

**Authors:** Hieu Van Dong, Giang Thi Huong Tran, Huong Thi Thu Nguyen, Tuong Manh Nguyen, Dai Quang Trinh, Van Phan Le, Kiattawee Choowongkomon, Jatuporn Rattanasrisomporn

**Affiliations:** 1Center for Advanced Studies for Agriculture and Food, Kasetsart University Institute for Advanced Studies, Kasetsart University, Bangkok 10900, Thailand; 2Faculty of Veterinary Medicine, Vietnam National University of Agriculture, Trau Quy Town, Gia Lam District, Hanoi 131000, Vietnam; 3Central Veterinary Medicine JSC No. 5, Ha Binh Phuong Industrial Zone, Hanoi 100000, Vietnam; 4Department of Biochemistry, Faculty of Science, Kasetsart University, Bangkok 10900, Thailand; 5Department of Companion Animal Clinical Science, Faculty of Veterinary Medicine, Kasetsart University, Bangkok 10900, Thailand

**Keywords:** NGPV, waterfowl parvovirus, Vietnam

## Abstract

**Simple Summary:**

Waterfowl parvoviruses are highly contagious lethal pathogens that cause economic loss in the duck production industry. In Vietnam, information is limited on epidemiological and genetic characterization of the viral genome. This study investigated epidemiological characteristics and the further genetic characterization of the viral genome in ducks farmed in northern Vietnam. The results revealed that waterfowl parvovirus was circulating among ducks at a moderately positive rate. The viral strains were clustered with another novel goose parvovirus from China.

**Abstract:**

In total, 130 tissue-pooled samples collected from ducks in some provinces/cities in north Vietnam were examined for waterfowl parvovirus genome identification. Twenty-six (20%) samples were positive for the parvovirus infection, based on polymerase chain reaction analysis. Of the 38 farms tested, 14 (36.84%) were positive for the waterfowl parvovirus genome. The rate of the parvovirus genome detection in ducks aged 2–4 weeks (37.04%) was significantly (*p* < 0.05) higher than that at ages <2 weeks (9.09%) and >4 weeks (16.30%). The positive rate on medium-scale farms (9.36%) was significantly (*p* < 0.05) lower than for small-scale (31.03%) and large-scale (29.73%) farms. The lengths of the four Vietnamese waterfowl parvovirus genomes identified were 4750 nucleotides. Among the four Vietnamese parvovirus genomes, nucleotide identities were from 99.29% to 99.87%. Phylogenetic analysis of the near-complete genomes indicated that the waterfowl circulating in northern Vietnam belonged to the novel goose parvovirus (NGPV) group. The Vietnamese NGPV group was closely related to the Chinese group. Recombination analysis suggested that the Vietnam/VNUA-26/2021 strain was generated by a recombination event. One positive selection site of the capsid protein was detected.

## 1. Introduction

Waterfowl parvoviruses are highly contagious, lethal pathogens for goslings and ducklings. Waterfowl parvoviruses belong to the species *Anseriform dependoparvovirus 1* of the genus *Dependoparvovirus* within the *Parvoviridae* family. The genus *Dependoparvovirus* can be divided into two divergent groups, consisting of Muscovy duck parvovirus (MDPV)-related groups and goose parvovirus (GPV)-related groups [1]. GPV, the agent of Derzy’s disease, causes the disease in young geese and Muscovy ducks with mortality rates up to 90% [2], while MDPV causes high mortality, watery diarrhea, wheezing, locomotor dysfunction, and stunning in Muscovy ducks [3]. Both GPV and MDPV can show 70–100% morbidity and mortality during the first 3–4 weeks of age. 

Waterfowl parvovirus is characterized as a non-enveloped virus. The viral genome consists of a single-strand DNA with around 5100 bp. The translated regions contain two open reading frames (ORFs). The left ORF encodes for the non-structural protein, while the remaining ORF encodes for capsid proteins VP1, VP2, and VP3 [4,5]. Both GPV and MDPV are antigenically related to each other as they share about 85% protein sequence homology [6]. In detail, they show nucleotide differences in VP1 of about 20–24%, whereas these differences in VP1 within the GPV and MDPV groups are only 0.1–7% and 0.1–1.9%, respectively [6,7,8,9]. 

It was reported that recombination events occurred among parvoviruses. As a result, a novel strain may be generated from the recombination event between vaccine and wild-type strains [10]. In 2015, Chen et al. found a novel goose parvovirus (NGPV) causing a severe disease in duck flocks in China [11]. Later, other studies noted that NGPV strains induced short beak and dwarfism syndrome (SBDS) in China, Egypt, and Poland [12,13,14]. NGPV viruses were considered to derive from classical GPV strains. Furthermore, a novel recombinant MDPV (rMDPV) was reported in ducks from several provinces in mainland China, where it caused high mortality and embolism in the intestinal tracts of infected ducklings aged less than 3 weeks [15,16,17]. 

In Vietnam in 2019, waterfowl parvoviruses have been reported in sick ducks with short beak and dwarfism syndrome. The detected waterfowl parvovirus was genetically grouped with the NGPV group based on the partial NS and VP1 genes [18]. However, information on waterfowl parvoviruses has been limited up to now. Therefore, we investigated infections of waterfowl parvoviruses in ducks farmed in some provinces of northern Vietnam. The current study also carried out molecular characterization of the near-complete waterfowl parvovirus genome. 

## 2. Materials and Methods

### 2.1. Ethics Statement

This research did not contain any studies involving human participants. Collection of duck tissue samples was conducted by the Vietnam National University of Agriculture after receiving institutional approval and permission from the owners of the ducks.

### 2.2. Samples

Tissue samples (brain, lung, liver, kidney, and Fabricius) from 130 broiler ducks aged 2–7 weeks were obtained from Thainguyen, Bacgiang, Haiduong, Thaibinh, Hungyen, and Hanoi in 2021 (Figure 1) and used in this study. From each flock, two to six diseased ducks were collected by the local veterinarian and transferred to the Vietnam National University of Agriculture for further analysis. The parvovirus vaccine was not used in tested duck farms. From each duck, collection of pooled tissue samples was performed. A 10% homogenate was prepared in phosphate-buffered saline.

### 2.3. DNA Extraction and Polymerase Chain Reaction (PCR)

DNA was extracted from the homogenized samples using Viral Gene-spin™ Viral DNA/RNA Extraction Kits (iNtRON Biotechnology, Seoul, Korea) according to the manufacturer’s instruction. Primers, PV-F, and PV-R were used to amplify the target 537 bp of the partial replication (Rep) geneor the parvovirus genome identification (Table 1), as previously described [4]. The thermal conditions were: 95 °C for 5 min, 35 cycles of 95 °C for 30 s, 55 °C for 30 s, 72 °C for 30 s, and extended at 72 °C for 10 min. A 1.5% agarose gel was used to run the PCR product. The product was observed by UV light. 

### 2.4. Nucleotide Sequencing and Phylogenetic Analyses

Five pairs of primers, GPV-P1-F/R, GPV-P2-F/R, GPV-P3-F/R, GPV-P4-F/R, and GPV-P5-F/R (Table 1), were used to amplify the near-whole genome sequence of the waterfowl parvovirus strains, as described previously [14]. Electrophoresis was performed with 1.5% agarose gel to separate the PCR products. Regarding the purification of PCR products, GeneClean^®^ II Kits (MP Biomedicals, Santa Ana, CA, USA) were used. Sequencing of the waterfowl parvovirus strains was demonstrated by 1st BASE, Malaysia. 

The Clustal W multiple alignment tool [19] in BioEdit v.7.2.5 [20] was used to align and analyze the nucleotide sequences and deduced aa sequences derived from GPV. Homology in nucleotide and aa sequences was examined using the GENETYX v.10 software (GENETYX Corp., Tokyo, Japan) and compared with other publicly available sequences using the BLAST program. A maximum likelihood method with the Hasegawa–Kishino–Yano model of nucleotide substitutions was used to construct the phylogenetic tree based on nucleotide sequences of 4 currently identified Vietnamese and 40 foreign waterfowl parvovirus strains (Table 2). The confidence values on phylogenetic trees were assessed based on bootstrapping with 1000 replicates using the MEGA6 6.06 version [21]. The near-whole genomes of the current waterfowl parvovirus strains were submitted to GenBank with accession numbers OP265005–OP265008.

### 2.5. Analyses of Recombination and Natural Selection Profiles

The current and other waterfowl parvovirus sequences available in GenBank were determined to be recombination events using the Recombination Detection Program (RDP) version Beta 4.97 [22] (including RDP, GENECONV, BootScan, MaxChi, Chimaera, SiScan, Phyl-Pro, LARD, and 3Seq methods) with default settings. An algorithm producing a result that had *p* value lower 0.05 was regarded as dependable. The identification of sites under positive or negative selection was evaluated by following a FUBAR (a Fast Unconstrained Bayesian AppRoximation) method (accessed on 1 July 2022) [23].

### 2.6. Statistical Analysis

Fisher’s exact test was used to identify significant differences in the rate of waterfowl parvovirus genome detection between geographical regions, age, or flock-size groups. A value of *p* < 0.05 was considered statistically significant.

## 3. Results

### 3.1. Detection of Waterfowl Parvovirus Genome in Field Samples

Of the 130 samples tested, 26 (20%) were positive for the waterfowl parvovirus genome based on the PCR method. Among the provinces/cities, Haiduong had the highest rate (33.33%), followed by (*p* > 0.05) Thainguyen (26.67%), Hanoi (22.92%), and Hungyen (20%), that were all significantly higher (*p* < 0.05) than that of Thaibinh (14.29%) and Bacgiang (13.33%) (Table 3). Of the 38 farms tested, 14 were positive for the waterfowl parvovirus genome (Table 3). 

The rate of parvovirus-positive ducks at age 2–4 weeks was 37.04% (10/27), which was highly significantly greater those aged <2 weeks (9.09%; *p* < 0.01) and >4 weeks (16.30%; *p* = 0.01) (Table 4). The flock levels were categorized as level 1 (number of ducks lower than 500, small scale), level 2 (number of ducks ranging 500–1000, medium scale), or level 3 (number of ducks more than 1000, large scale). The rates of parvovirus detection for the level 1 and level 3 flocks were 31.03% and 29.73%, respectively, which were significant (*p* < 0.05) higher than for level 2 flocks (9.38%), as shown in Table 4.

### 3.2. Characterization of Waterfowl Parvovirus Genome and Protein

The near-complete genome of the four Vietnamese waterfowl parvoviruses obtained in this study was 4750 nucleotides in length. The viral genomes of the four Vietnamese waterfowl parvovirus types consisted of two ORFs. Neither deletion nor insertion mutations were detected in the translated region of the current parvovirus.

Regarding genetic analysis, four parvovirus-detected samples obtained at different locations were randomly determined for viral genome sequencing. The alignment and comparison of the near-whole genome (4750 bp) of the four current strains and other reported sequences from GenBank were performed. The nucleotide identity ranged from 99.29% to 99.87% among the four Vietnamese waterfowl parvovirus strains obtained in this study. Among these, the highest nucleotide identity was between Vietnam/VNUA-07/2021 and Vietnam/VNUA-26/2021 (99.87%) while the lowest was between Vietnam/VNUA-30/2021 and Vietnam/VNUA-94/2021 (99.29%). Comparing the viral genomes from the four Vietnamese parvovirus strains in this study and those sequences abroad, four Vietnamese strains shared nucleotide identity of 99.17% (Vietnam/VNUA-94/2021 versus China/HuN001), 99.04% (Vietnam/VNUA-07/2021 versus China/HuN001, Vietnam/VNUA-26/2021 versus China/HuN001), and 98.82% (Vietnam/VNUA-30/2021 versus China/HuN001).

Phylogenetic analysis based on the near-complete genome (4880 bp), full-length Cap (2199 bp), and full-length Rep (1884 bp) gene sequences indicated that the current Vietnamese parvovirus strains belonged to the NGPV group. The four Vietnamese parvovirus strains obtained differed from vaccine strains Taiwan/VG32/1/2008 (EU583392.1), Taiwan/82-0321V/1982 (EU583389.1), and China/FZ91-30/1991 (KT865605.1). The present Vietnamese parvovirus strains were clustered with those of viral strains from China (Figure 2, Figure 3 and Figure 4).

Deduced aa sequences of the Rep and Cap proteins of the Vietnamese NGPV strains were compared with other waterfowl strains in various lineages. Four and fourteen major variable amino acid substitutions were detected on Rep and Cap proteins, respectively, of the Vietnamese NGPV strains in the current study, compared to other GPV and MDPV viruses (Table 5 and Table 6).

### 3.3. Recombination Analysis

Recombination analyses of the full-length Cap gene sequences suggested that Vietnam/VNUA-26/2021 resulted from a recombination event. The major and minor parents were the Duck/China/GXN45/2017 NGPV and Goose/China/YZ/2013 GPV viruses. This putative recombination event was detected by seven out of the nine methods using RDP 4 (Table 7). The two breakpoints were detected and located at residues 112 and 620 of the Cap gene sequence (Figure 5). No recombination events were detected in the Vietnamese NGPV Rep gene and the untranslated regions of the waterfowl parvovirus genome (data not shown).

### 3.4. Evolutionary Analysis of Viral Genome

Analyses of the natural selection profiles of the Vietnamese NGPV sequences indicated that eight residue sites (137, 147, 157, 159, 161, 168, 472, and 686) of the Cap protein and one residue (508) site of the Rep protein were from negative selection. However, only one positive selection was found at the residue 507 site of the Cap protein (Table 8).

## 4. Discussion

Understanding waterfowl parvovirus infections is important as they are associated with SBDS in ducks with a high or low mortality [12,13,14]. Recently, waterfowl parvoviruses were characterized based on partial sequences of the VP1 and NS1 gene sequences and divided into the NGPV group in Hungyen province [18]. The Vietnamese waterfowl parvovirus genome sequences are still limited in GenBank; additional sequence data of the complete genome are necessary to further characterize and understand the evolution of viral strains. This was the first study to conduct epidemiological analysis and to further characterize the whole-genome sequences (4880 bp) of Vietnamese waterfowl parvovirus strains.

In the current study, the waterfowl parvovirus genome was detected in 20% of duck samples, which was lower than that of ducklings from outbreak farms with SBDS in tissue samples (33/33 (100%)) or with no clinical signs (7/12 (58.33%)) [24]. The differences in the sampling numbers, location, time, and sensitivity of the detection methods were considered a reasonable cause for the variation in the result. Furthermore, 36.84% of the duck farms were positive for the viral genome in northern provinces in Vietnam. The current results suggested that waterfowl parvovirus may circulate and affect duck production in Vietnam. In the current study, we first found a moderate waterfowl parvovirus genome frequency of detection in Vietnamese ducks at age 2–4 weeks (37.04%), with the frequency of detection of viral genome higher than that in younger ducks aged <2 weeks (9.09%) and older ducks aged >4 weeks (16.30%). These findings suggested that during age 2–4 weeks the ducklings may be susceptible to viral infection.

Genetic and phylogenetic analyses indicated that the Vietnamese and Chinese strains were clustered in a single genogroup belonging to NGPV (Figure 2, Figure 3 and Figure 4). The results suggested that these viruses may have a similar origin, or that these viruses were widespread in Asia. In addition, the classification of the Vietnamese NGPV sequences was similar based on the complete genome, Rep gene, and Cap gene sequences, suggesting that using each gene sequence could characterize the viral strains. Amino acid substitutions on Rep and Cap proteins play critical roles in the change of host range and pathogenicity. In terms of Rep protein, a previous study reported that substitution at residue 140 (A to S), which was not be found in the current four Vietnamese strains, was associated with viral transcription [25]. Yu et al. noted that five substitutions at residues 494, 553, 555, 573, and 594 on Rep protein play roles in the antigenic epitopes of B cells [26]. Only one substitution at residue 555 N/D to P/T was detected in the present study. The roles of the remaining three substitutions on Rep protein of the Vietnamese strains were still unclear. In this study, 14 substitutions on Cap protein did not match with residues 35 and 660, which may be related to the host range identification of NGPV [27]. 

SBDS was recognized in the NGPV group in China in 2015 [11]. Since then, this pathogen has been critical in causing disease in ducks, mainly in China but also in other countries. In the current study, the four NGPVs were detected based on their complete genome sequences. The four Vietnamese NGPVs obtained were closely related to NGPV in China in 2019 (HuN18). The HuN18 NGPV virus has been isolated from ducks with BADS in China [28]. Therefore, it could be speculated that the current Vietnamese NGPV strains may be high-virulence strains. 

Recombination has been reported in the evolutionary processes of waterfowl parvoviruses and described [17,28,29,30] based on the complete genome sequences. A recombination event may generate a novel waterfowl parvovirus strain [10]. The current study revealed the first evidence supporting a recombination event in the Vietnamese NGPV strain in the protein-coding region of the Capsid gene. In general, recombination usually occurs between viruses that are located closely to each other. However, the major and minor parents were from China. The explanation for this situation is that Vietnam and China share thousands of kilometers along their border. The recombination event could have occurred elsewhere in China and then could have been transferred to Vietnam through daily trading of live birds along the border between Vietnam and China [31]. Recombination may occur between the Vietnamese waterfowl parvovirus strains, which might be detected if there are sufficient data. Analysis of the additional viral genomes should be further studied.

Another evolutionary process is natural selection. Fan et al. [32] reported nine sites on VP protein that were estimated as positively selected sites, that may have been associated with host range [32]. In the current study, only one positive selection on a VP protein was found (residue 507), which has not been characterized. Further studies should be conducted to clarify this point.

## 5. Conclusions

A high positive rate of waterfowl parvovirus was detected among young ducks, aged 3–4 weeks, in flocks in northern Vietnam. The current study revealed that the four Vietnamese waterfowl parvovirus strains belonged to the NGPV group, based on phylogenetic and molecular analyses of the whole genome. The characterization of NGPV strains circulating in Vietnam was first reported based on viral whole-genome sequences. This study also detected a recombination event in the Vietnamese NGPV strain. Only one aa sequence on a Cap protein was identified under positive selection.

## Figures and Tables

**Figure 1 animals-12-02846-f001:**
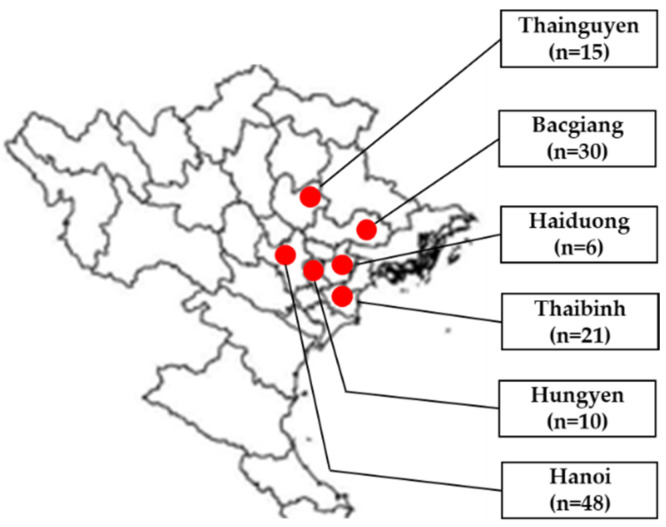
Geographical locations of sample areas in northern Vietnam (red circles) and distribution of sampling size in Thainguyen (15), Bacgiang (30), Haiduong (6), Thaibinh (21), Hungyen (10), and Hanoi (48).

**Figure 2 animals-12-02846-f002:**
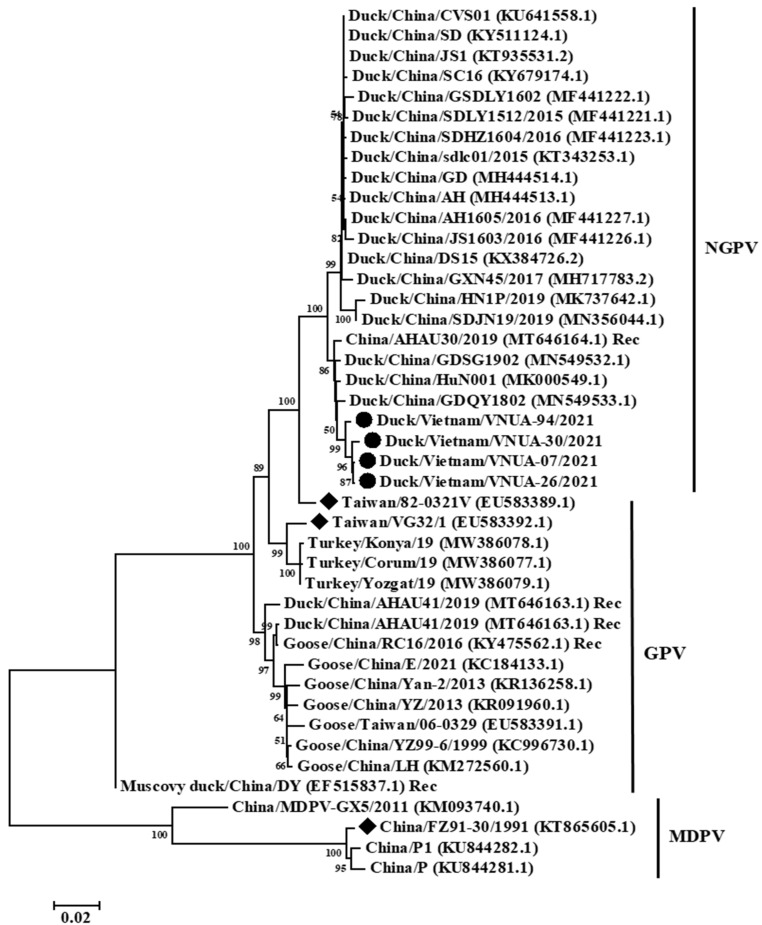
Phylogenetic trees of the near-complete genome (4750 bp) sequences of Vietnamese parvovirus strains compared with those available in GenBank. GenBank sequences are indicated by the country name/accession number. The maximum likelihood method in the MEGA6 software was used to establish phylogenetic trees (1000 bootstrap replicates). Numbers at each branch point indicate bootstrap values ≥50% in the bootstrap interior branch test. The current Vietnamese and vaccine strains are indicated by circles and diamonds, respectively.

**Figure 3 animals-12-02846-f003:**
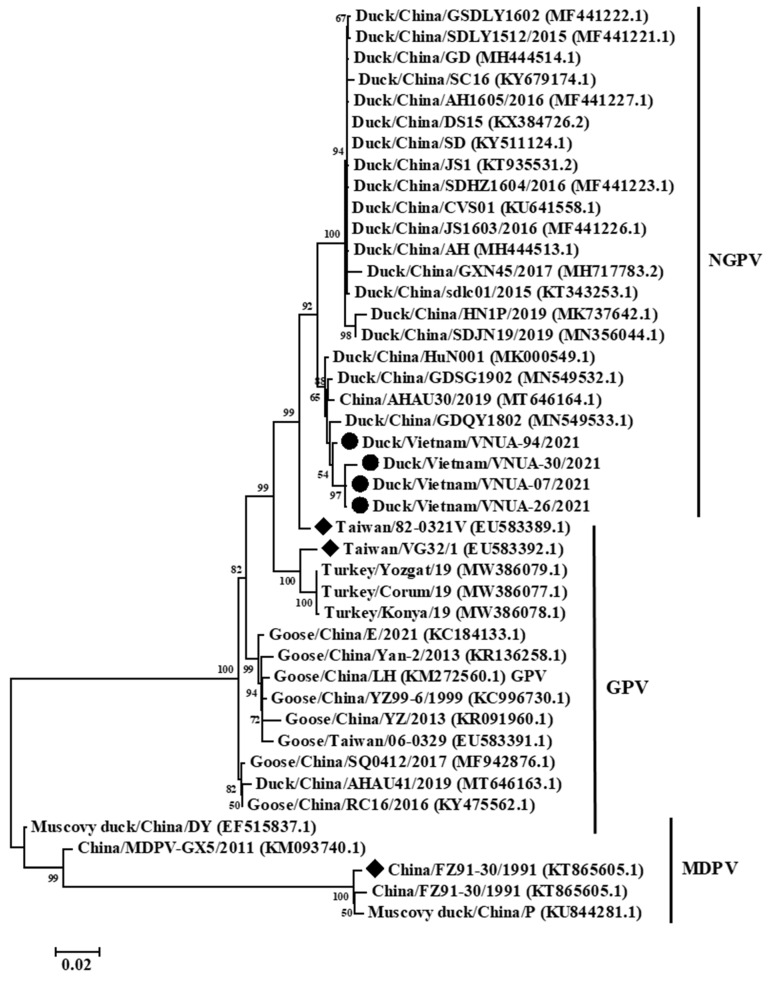
Phylogenetic trees of the full-length Cap gene (2199 bp) sequences of Vietnamese parvovirus strains compared with those available in GenBank. GenBank sequences are indicated by the country name/accession number. The maximum likelihood method in the MEGA6 software was used to establish phylogenetic trees (1000 bootstrap replicates). Numbers at each branch point indicate bootstrap values ≥50% in the bootstrap interior branch test. The current Vi-etnamese and vaccine strains are indicated by circles and diamonds, respectively.

**Figure 4 animals-12-02846-f004:**
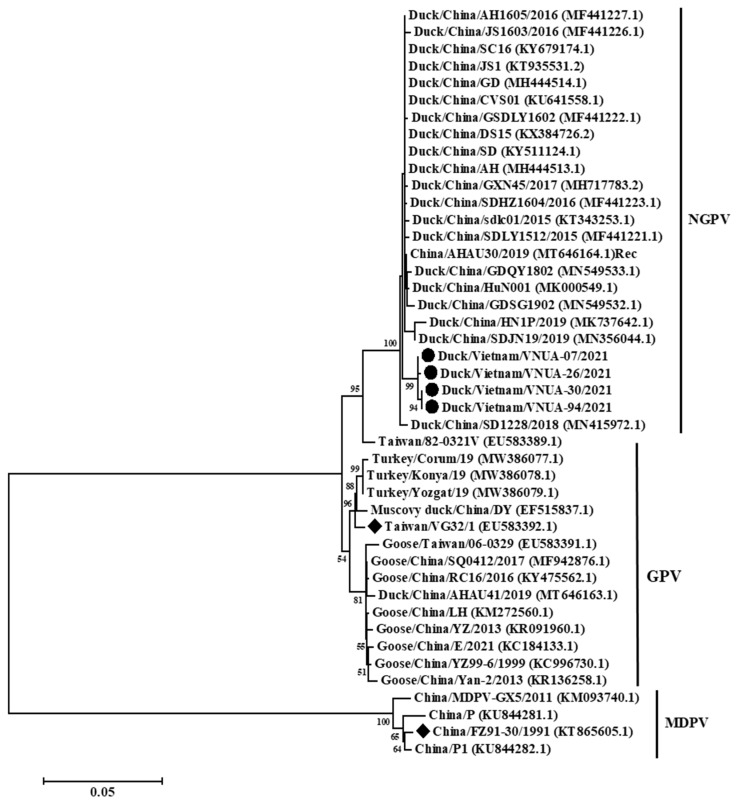
Phylogenetic trees of the full-length Rep gene (1884 bp) sequences of Vietnamese parvovirus strains com-pared with those available in GenBank. GenBank sequences are indicated by the country name/accession number. The maximum likelihood method in the MEGA6 software was used to establish phylogenetic trees (1000 bootstrap repli-cates). Numbers at each branch point indicate bootstrap values ≥50% in the bootstrap interior branch test. The current Vietnamese and vaccine strains are indicated by circles and diamonds, respectively.

**Figure 5 animals-12-02846-f005:**
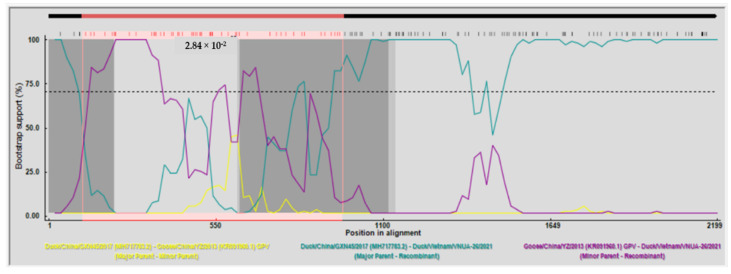
Detection of recombination events using BootScan analysis of full-length Cap gene sequences of Vietnam/VNUA-26/2021 NGPV strains. The pairwise distance model with window size 200, step size 20, and 1000 bootstrap replicates was generated by the RDP 4 program.

**Table 1 animals-12-02846-t001:** Primers used in this study.

Target	Name	Sequence (5’–3’)	PCR Product (bp)	Location of Target Genes on the Viral Genome	Reference
PCR	PV-F	CCAAGCTACAACAACCACAT	539		[4]
	PV-R	TGAGCGAACATGCTATGGAAGG			
Sequencing	GPV-P1-F	CTTATTGGAGGGTTCGTTCGT	176	5′-UTR	[14]
	GPV-P1-R	GCATGCGCGTGGTCAACCTAACA			
	GPV-P2-F	GCATGCCGCGCGGTCAGCCCAATA	1033	5′-UTR/Rep	[14]
	GPV-P2-R	ATTTCAATGAGCCAATCAACAAGG			
	GPV-P3-F	GCCTTTATTTACTGCTGC	1443	Rep	[14]
	GPV-P3-R	GCTTTCAGATTCCGCCAC			
	GPV-P4-F	CTTGATGATGCTGAAAATGAAC	1446	Rep/Cap	[14]
	GPV-P4-R	GCCCATGGTGCCATAAGC			
	GPV-P5-F	CGCTCATTCACAGGACTTAGACA	1170	Cap/3′-UTR	[14]
	GPV-P5-R	GCATGCGCGTGGTCAACCTAACA			

**Table 2 animals-12-02846-t002:** Description of waterfowl parvovirus isolates compared in this study.

GenBank Accession No.	Strain	Location	Source	Year
MN549533.1	GDQY1802	China	Duck	2018
MN549532.1	GDSG1902	China	Duck	2018
KT935531.2	JS1	China	Duck	2015
KX384726.2	DS15	China	Duck	2015
KU641558.1	CVS01	China	Duck	2015
KY679174.1	SC16	China	Duck	2016
MF441227.1	AH1605	China	Duck	2016
MH444513.1	AH	China	Duck	2018
MN415972.1	SD1228	China	Duck	2018
MN415972.1	SD1228	China	Duck	2018
MN356044.1	SDJN19	China	Duck	2019
EU583392.1	VG32/1 vaccine	Taiwan	Goose	2008
EU583389.1	82-0321V vaccine	Taiwan	Goose	1982
MT646164.1	AHAU30	China	Duck	2019
EF515837.1	DY	China	Muscovy duck	2007
MT646163.1	AHAU41	China	Duck	2019
MF942876.1	SQ0412	China	Goose	2017
KY475562.1	RC16	China	Goose	2016
KC996730.1	YZ99-6	China	Goose	1999
KM272560.1	LH	China	Goose	2012
KR136258.1	Yan-2	China	Goose	2013
KR091960.1	YZ	China	Goose	2013
EU583391.1	06-0329	Taiwan	Goose	2008
KC184133.1	E	China	Goose	2021
MW386077.1	Corum/19	Turkey	Goose	2019
MW386079.1	Yozgat/19	Turkey	Goose	2019
MW386078.1	Konya/19	Turkey	Goose	2019
MF441223.1	SDHZ1604	China	Duck	2016
MF441221.1	SDLY1512	China	Duck	2015
MF441226.1	JS1603	China	Duck	2016
MF441222.1	SDLY1602	China	Duck	2016
MH444514.1	GD	China	Duck	2016
MK000549.1	HuN001	China	Duck	2018
MK737642.1	HN1P	China	Duck	2019
KY511124.1	SD	China	Duck	2015
KT343253.1	sdlc01	China	Duck	2015
KM093740.1	MDPV-GX5	China	Muscovy duck	2011
KU844282.1	P1	China	Muscovy duck	2016
KT865605.1	FZ91-30 vaccine	China	Muscovy duck	1991
KU844281.1	P	China	Muscovy duck	1988

**Table 3 animals-12-02846-t003:** Detection of waterfowl parvovirus genome in field duck samples obtained from different northern provinces/cities of Vietnam.

Province/City	No. of Collected Samples	Gene-Positive Samples/(%)	No. of Flocks	Gene-Positive Flocks/(%)
Hanoi	48	11/(22.92) ^a,b^	11	4/(36.36) ^c^
Haiduong	6	2/(33.33) ^a,b^	2	1/(50.00) ^c,d^
Thainguyen	15	4/(26.67) ^a^	3	2/(66.67) ^d^
Bacgiang	30	4/(13.33) ^b^	11	3/(27.27) ^c,e^
Thaibinh	21	3/(14.29) ^b^	7	2/(28.57) ^c,e^
Hungyen	10	2/(20.00) ^a,b^	4	2/(50) ^c,d^
Total	130	26/(20.00)	38	14/(36.84)

Letters ^a,b,c,d,e^ indicate that the groups were significantly (*p* < 0.05) different from each other.

**Table 4 animals-12-02846-t004:** Detection of waterfowl parvovirus genome in field duck samples by age and flock size.

Criteria	Ages/Duck Heads	Samples (n)	Gene-Positive Samples/(%)
Age (weeks)	<2	11	1/(9.09) ^a^
	2–4	27	10/(37.04) ^b^
	>4	92	15/(16.30) ^a^
Flock size	<500	29	9/(31.03) ^a^
	500–1000	64	6/(9.38) ^b^
	>1000	37	11/(29.73) ^a^

Letters ^a,b^ indicate that the groups were significantly (*p* < 0.05) different from each other.

**Table 5 animals-12-02846-t005:** Amino acid substitutions in Rep protein of Vietnamese NGPV strains.

Virus Strain/Consensus ^a^	Sites in Rep Protein			
	154	551	555	560
GPV	T	R	N	C
MDPV	T	K	D	C
Vietnamese NGPVs	T/S	K/R	P/T	G/C

^a^ Deduced amino acid consensus was released with 102 waterfowl parvovirus sequences retrieved from GenBank by using GENETYX version 10.

**Table 6 animals-12-02846-t006:** Amino acid substitutions in Cap protein of Vietnamese NGPV strains.

Virus Strain/Consensus ^a^	Sites in Cap Protein													
	142	144	178	180	206	386	390	448	449	451	458	461	507	509
GPV	D	V	S	A	A	N	A	D	G	R	A	G	D	Q/E
MDPV	E	V	N	G	A	N	A	D/N	S/G	R	A	G	D	E
Vietnamese NGPVs	E/D	I/V	T/S	A/V	T/A	N/D	A/P	D/G	S/R	R/G	A/T	G/R	E/D/A	Q/P

^a^ Deduced amino acid consensus was released with 102 waterfowl parvovirus sequences retrieved from GenBank by using GENETYX version 10.

**Table 7 animals-12-02846-t007:** Recombination statistics of Vietnam/VNUA-26/2021 Cap gene sequence using RDP 4.

Method	Recombination *p*-Value
RDP	1.89 × 10^−2^
GENECONV	4.77 × 10^−4^
BootScan	2.84 × 10^−2^
MaxChi	2.72 × 10^−5^
Chimaera	8.11 × 10^−6^
SiScan	5.41 × 10^−12^
PhylPro	-
LARD	-
3Seq	4.48 × 10^−3^

Recombination events with *p*-value < 0.05 were regarded as reliable.

**Table 8 animals-12-02846-t008:** Natural selection profile in waterfowl parvovirus capsid and replication protein of four current Vietnamese NGPV strains.

Protein	Site	a	b	b–a	Prob [a > b]	Prob [a < b]	Bayes Factor [a < b]
Capsid	137	26.755	2.394	−24.361	0.912	0.063	0.083
	147	26.312	2.169	−24.142	0.916	0.06	0.078
	157	25.594	2.021	−23.573	0.915	0.061	0.079
	159	26.745	2.188	−24.557	0.917	0.06	0.079
	161	24.335	2.109	−22.226	0.906	0.068	0.089
	168	25.174	2.226	−22.948	0.908	0.065	0.086
	472	24.338	2.108	−22.23	0.906	0.068	0.089
	507	4.423	33.746	29.323	0.034	0.926	15.48
	686	25.729	2.158	−23.571	0.912	0.063	0.083
Replication	508	30.806	3.718	−27.088	0.905	0.062	0.077

## Data Availability

The data presented in this study are available within the article. Raw data supporting this study are available from the corresponding author.

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
