# Peer review of "Epidemiological Analysis and Genetic Characterization of Parvovirus in Ducks in Northern Vietnam Reveal Evidence of Recombination"

_animals, 2022, doi:10.3390/ani12202846_

Round 1
Reviewer 1 Report
This is an informative article about the epidemiology of waterfowl parvovirus in Vietnam, that goes dipper into analyze the almost complete sequence of 4 genomes. The article could be improved by a stronger discussion, on both the epidemiological and molecular aspects, as well as more in deep analysis of the phylogenetic studies. Do the viruses groups equally if the comparison if with the full genomes or the viral proteins for example?
The following specific comments should be also addressed to improve this article.
Table 1 contains the sequence of the primers that were used to amplify different fragments of the parvovirus genome that were send to sequences, although all are published primers, it would be really explicative if the authors could mention to which part of the genome, or a reference genome, each set of primers binds.
Table 3 indicates the number of samples collected and samples positive by PCR, but it is not explained what the flock number is referring to, Is it the number of farms form were the samples were collected? Th number of groups that were in each area in total, independent of the fact they were in the same farm? It is also not discussed how is the distribution of positive samples among the positive flocks. If as shown in table 4, the flocks with more than 500 animals have only 6-11 positive sample, that represent between 9.38-29.73 positive samples according to the number of samples collected, would this number of samplings be representative of a disseminated virus among the population? Is there correlation with pathology among a similar % of animals among all the population?
The authors describe a series of substitutions in several residues, but do not clarify with which virus are they comparing, or if those substitution are only present in the nearly complete genomes and differes from all other described waterfowl parvovirus. Also, table 5 is labeled wrongly since it indicates Cap, when it should be Rep. It would be interesting if the authors discussed were in the possible structure these substitutions are or if they predict to affect some functional domain of the protein. This last comment is valid for the Cap substitutions too.
If the authors are presenting the substitutions in a table, does not seem necessary to mention each specific one in the text.
Lines 216, does not state that the 14 substitutions mentioned corresponds to Cap.
Figure 2 could be divided in 3 independent figures in a bigger size; thus the reader can indeed read each of the elements presented on those trees. Also, a bit more information could be explained and discussed from those analysis.
Lines 282: Them..would they mean then?
Author Response
We are grateful to the reviewer for their valuable comments and helpful suggestions. We have revised the manuscript based on the reviewers’ comments. All changes in the revised manuscript are indicated by a red-font color. Below, we have provided our point-by-point responses to the reviewers’ comments.
Reviewer 1:
- This is an informative article about the epidemiology of waterfowl parvovirus in Vietnam, that goes dipper into analyze the almost complete sequence of 4 genomes. The article could be improved by a stronger discussion, on both the epidemiological and molecular aspects, as well as more in deep analysis of the phylogenetic studies. Do the viruses groups equally if the comparison if with the full genomes or the viral proteins for example?
We thank the reviewer’s comments. Based on the current results of full genomes and viral protein analysis, the phylogenetic division is similar among Vietnamese waterfowl parvoviruses.
Table 1 contains the sequence of the primers that were used to amplify different fragments of the parvovirus genome that were send to sequences, although all are published primers, it would be really explicative if the authors could mention to which part of the genome, or a reference genome, each set of primers binds.
We agree with the reviewer’s comments and suggestions. The changes have been made in Table 1 in the revised manuscript.
Table 3 indicates the number of samples collected and samples positive by PCR, but it is not explained what the flock number is referring to, Is it the number of farms form were the samples were collected? The number of groups that were in each area in total, independent of the fact they were in the same farm? It is also not discussed how is the distribution of positive samples among the positive flocks.
We thank for the reviewer’s comments and questions.
We apologize for our mistakes. We provided a sentence in lines 101–102 to describe how the samples were collected. From each farm, two to six diseased ducks were collected and sent to us for further analysis. In one farm, not all samples tested were positive for the viral infection, but this information seemed to be raw data and it is not statistical data.
If as shown in table 4, the flocks with more than 500 animals have only 6-11 positive sample, that represent between 9.38-29.73 positive samples according to the number of samples collected, would this number of samplings be representative of a disseminated virus among the population? Is there correlation with pathology among a similar % of animals among all the population?
We thank for the reviewer’s comments and suggestions. In this study, we started as a case study (positive rate), but not prevalence since the sample size was small in comparison to duck population. Based on the current results, we could not find any correlation with pathology among a similar % of animals among all the population.
The authors describe a series of substitutions in several residues, but do not clarify with which virus are they comparing, or if those substitution are only present in the nearly complete genomes and differs from all other described waterfowl parvovirus. Also, table 5 is labeled wrongly since it indicates Cap, when it should be Rep.
- We agree with the reviewer’s comments and suggestions. We added additional information “compared to consensus GPV and MDPV viruses” in lines 215, 244-245 in the revised manuscript.
- We apologize for the mistake. The change has been made in Table 5.
It would be interesting if the authors discussed were in the possible structure these substitutions are or if they predict to affect some functional domain of the protein. This last comment is valid for the Cap substitutions too.
We thank for the reviewer’s suggestions. The changes have been made in lines 303-313 in the revised manuscript.
- If the authors are presenting the substitutions in a table, does not seem necessary to mention each specific one in the text.
We thank for the reviewer’s suggestion. We deleted each specific substitution in the text.
Lines 216, does not state that the 14 substitutions mentioned corresponds to Cap.
We agree with the reviewer’s comment and suggestion. We have changed the sentence in lines 217-219 in the revised manuscript.
- Figure 2 could be divided in 3 independent figures in a bigger size; thus the reader can indeed read each of the elements presented on those trees. Also, a bit more information could be explained and discussed from those analysis.
We thank for the reviewer’s comments and suggestions. Additional figures have been added and we discussed more in lines 298-304 in the revised manuscript.
Lines 282: Them..would they mean then?
We thank for the reviewer’s correction. The change has been made.
Additional change:
Some minor changes made were also highlighted in red color.

Reviewer 2 Report
The paper describes the detection of waterfowl parvoviruses in Vietnamese duck flocks. The authors recover genome sequences of waterfowl parvoviruses and perform genomic analysis. It is important to study the emergence of these apparently novel parvoviral pathogens. The paper takes a standard, descriptive approach that is generally speaking appropriate for this kind of study. I only have minor comments.
Table 7 - please explain the meaning of the column headers.
Line 276-277 - I think the authors mean frequency of detection - not 'rate'
Line 392 - Parvoviruses not 'pervoviruses'
Author Response
We are grateful to the reviewer for their valuable comments and helpful suggestions. We have revised the manuscript based on the reviewers’ comments. All changes in the revised manuscript are indicated by a red-font color. Below, we have provided our point-by-point responses to the reviewers’ comments.
Reviewer 2: The paper describes the detection of waterfowl parvoviruses in Vietnamese duck flocks. The authors recover genome sequences of waterfowl parvoviruses and perform genomic analysis. It is important to study the emergence of these apparently novel parvoviral pathogens. The paper takes a standard, descriptive approach that is generally speaking appropriate for this kind of study. I only have minor comments.
We thank for the reviewer’s comments on our original manuscript.
- Table 7 - please explain the meaning of the column headers.
We thank for the reviewer’s question. About the column headers, a and b are probability of negative and positive selections, respectively. The BayesFactor factor is a ratio of two competing statistical models represented by their marginal likelihood, and is used to quantify the support for one model over the other. In this case, site 507 of Cap protein indicated with BayesFactor = 15.48 which mean the probability of the positive selection is high.
- Line 276-277 - I think the authors mean frequency of detection - not 'rate'
We thank for the reviewer’s suggestion. The change have been made in the revised manuscript.
- Line 392 - Parvoviruses not 'pervoviruses'
We agree with the reviewer’s suggestion. The change has been made in the revised manuscript.
Additional change:
Some minor changes made were also highlighted in red color.
